# Familial Hypercholesterolemia in Premature Acute Coronary Syndrome. Insights from CholeSTEMI Registry

**DOI:** 10.3390/jcm9113489

**Published:** 2020-10-29

**Authors:** Rebeca Lorca, Andrea Aparicio, Elias Cuesta-Llavona, Isaac Pascual, Alejandro Junco, Sergio Hevia, Francisco Villazón, Daniel Hernandez-Vaquero, Jose Julian Rodríguez Reguero, Cesar Moris, Eliecer Coto, Juan Gómez, Pablo Avanzas

**Affiliations:** 1Reference Unit of Familiar Cardiomyopathies-HUCA, Área del Corazón y Departamento de Genética Molecular, Hospital Universitario Central Asturias, 33014 Oviedo, Spain; lorcarebeca@gmail.com (R.L.); eliascllavona@gmail.com (E.C.-L.); josejucasa@yahoo.es (J.J.R.R.); cesarmoris@gmail.com (C.M.); eliecer.coto@sespa.es (E.C.); uo167835@uniovi.es (J.G.); 2Heart Area, Hospital Universitario Central de Asturias, 33014 Oviedo, Spain; apariciogavilanes@gmail.com (A.A.); ajuncovicente@gmail.com (A.J.); josesergiohevia@gmail.com (S.H.); dhvaquero@gmail.com (D.H.-V.); avanzas@gmail.com (P.A.); 3Instituto de Investigación Sanitaria del Principado de Asturias, ISPA, 33014 Oviedo, Spain; 4Endocrinology Department, Hospital Universitario Central Asturias, 33014 Oviedo, Spain; fvillazon@yahoo.es

**Keywords:** familial hypercholesterolemia (FH), myocardial infarction with ST elevation (STEMI), premature coronary artery disease (CAD), Dutch Lipid Clinic Network (DLCN)

## Abstract

Familial hypercholesterolemia (FH) is an underdiagnosed genetic inherited condition that may lead to premature coronary artery disease (CAD). FH has an estimated prevalence in the general population of about 1:313. However, its prevalence in patients with premature STEMI (ST-elevation myocardial infarction) has not been widely studied. This study aimed to evaluate the prevalence of FH in patients with premature STEMI. Cardiovascular risk factors, LDLc (low-density lipoprotein cholesterol) evolution, and differences between genders were also evaluated. Consecutive patients were referred for cardiac catheterization to our center due to STEMI suspicion in 2018. From the 80 patients with confirmed premature CAD (men < 55 and women < 60 years old with confirmed CAD), 56 (48 men and eight women) accepted to be NGS sequenced for the main FH genes. Clinical information and DLCN (Dutch Lipid Clinic Network) score were analyzed. Only one male patient had probable FH (6–7 points) and no one reached a clinically definite diagnosis. Genetic testing confirmed that the only patient with a DLCN score ≥6 has HF (1.8%). Smoking and high BMI the most frequent cardiovascular risk factors (>80%). Despite high doses of statins being expected to reduce LDLc levels at STEMI to current dyslipidemia guidelines LDL targets (<55 mg/dL), LDLc control levels were out of range. Although still 5.4 times higher than in general population, the prevalence of FH in premature CAD is still low (1.8%). To improve the genetic yield, genetic screening may be considered among patients with probable or definite FH according to clinical criteria. The classical cardiovascular risk factors prevalence far exceeds FH prevalence in patients with premature STEMI. LDLc control levels after STEMI were out range, despite intensive hypolipemiant treatment. These findings reinforce the need for more aggressive preventive strategies in the young and for intensive lipid-lowering therapy in secondary prevention.

## 1. Introduction

Coronary artery disease (CAD) is not only the most frequent cause of death worldwide, but also one of the most frequent causes of premature death, with an increasing prevalence [1]. In fact, reported disability-adjusted life years (DALYs) due to CAD is 8.6 per 1000 habitants, 96% of them corresponding to years of life lost (YLLs) [2]. Further, between the ages of 30 and 54, ischemic heart disease has an incidence of 1% per year in men and 0.4% in women [3].

In this context, familial hypercholesterolemia (FH) is known to be a predisposing cause of premature CAD [4]. Patients with this genetic disorder have elevated low-density lipoprotein cholesterol (LDLc) levels from birth. This high LDLc levels leads to an accelerated atherosclerosis [5] that consequently predisposes to early CAD [4]. Moreover, FH is considered not only one of the most common monogenic conditions [5], but also the genetic disorder most frequently found in ischemic heart disease [4]. Pathogenic variants in the genes encoding for the LDL receptor (*LDLR*), apolipoprotein B (*APOB*) and proprotein convertase subtilisin/kexin type 9 (*PCSK9)*, are known to cause heterozygous FH, an autosomal dominant inherited condition [4,5]. While heterozygous FH is quite common, homozygous FH is very rare. Pathogenic variants in LDL-R adaptor protein 1 gene (*LDLRAP1)* can produce FH with an autosomal recessive inherited pattern. In general, most pathogenic variants in FH are identified in *LDLR* gene [4], accounting for more than 95% of the cases [6]. On the other hand, other interesting genes, not responsible for FH but associated with hyperlipidemia (such as *APOE* and *LIPA*), have been identified.

There are different clinical tools and criteria available for FH diagnosis [6], with the Dutch Lipid Clinic Network (DLCN) criteria being the most frequently used in Europe [7,8]. DLCN criteria are based on untreated LDLc levels, family history of premature cardiovascular disease (CVD) in first degree relatives, their own CVD history and physical examination findings such as tendon xanthomas or corneal arch, among other parameters [7,8]. Nevertheless, despite the value of these clinical diagnostic criteria, genetic testing is the gold standard for FH diagnosis [4,9]. A total score ≥8 makes the FH diagnosis definite. Therefore, identifying a pathogenic variant in genetic testing corresponds to 8 points. FH is considered unlikely with punctuations <3; “possible” between 3–5; and “probable” between 6–8 points.

In this regard, an early diagnosis of FH seems vital in order to personalize the medical treatment. Since effective LDLc lowering therapies are available; morbidity and mortality have substantially decreased. In fact, it has been proposed that intensive treatment with statins may reduce the cardiovascular risk of FH patients to the general population [4]. In current dyslipidemia guidelines [10], LDL target level has lower from 70 mg/dL to 55 mg/dL in CAD secondary prevention. This recommendation is extensible for FH patients plus atherosclerotic CVD or another major risk factor [10]. Statins are the first-choice therapy, combined with other lipid-lowering meditation if needed [6]. Indeed, PCSK9-inhibitors are now also available for these patients with very high cardiovascular risk [11].

However, despite its implication in CVD, FH is still an underdiagnosed condition. Although its prevalence was believed to be around 1 in every 500 people, recent data suggest that it could be closer to one in every 200–313 people [12,13,14,15,16]. Nonetheless, it has been said that only 1% of patients with FH have been correctly identified [5]. In this respect, the prevalence of FH evaluated by genetic testing among patients with premature atherothrombotic CAD, presenting as myocardial infarction (MI) factors has not been widely examined.

In this study, we aimed to explore the prevalence of a genetic condition (FH) within a selected population with confirmed atherothrombotic CAD. We systematically evaluated the prevalence of FH, evaluated by both clinical and genetic testing, in a cohort of patients with confirmed premature atherothrombotic CAD presenting with MI with ST-elevation (STEMI). Known cardiovascular risk factors and cholesterol levels (at STEMI, prior to STEMI, and after STEMI) were also evaluated.

## 2. Materials and Methods

### 2.1. General Definitions

There are 3 traditional types of acute coronary syndrome (ACS): ST-elevation myocardial infarction (STEMI), Non-ST-elevation myocardial infarction (NSTEMI), and unstable angina. Acute MI is defined as acute myocardial injury detected by abnormal cardiac biomarkers in the setting of evidence of acute myocardial ischaemia [17]. MI is classified in 5 types, based on the etiology and pathogenesis [17]. MI caused by atherothrombotic CAD and usually precipitated by atherosclerotic plaque disruption is designated as a type 1 MI. However, in type 2 MI there is demand-supply mismatch resulting in myocardial ischemia [17]. This demand/supply mismatch can be due to multiple reasons, including but not limited to the presence of coronary obstruction. Mechanisms include hypoxemia hypo/hypertension, arrhythmias, anemia, and also vasospasm, acute aortic dissection coronary embolus, spontaneous coronary artery dissection, etc., [17].

Patients with acute chest pain and persistent ST-segment elevation on a surface electrocardiogram (generally reflects an acute total or subtotal coronary occlusion) ultimately develop STEMI [18]. Differential Diagnosis includes other pathologies that can cause ST-segment elevations without significant coronary artery disease, such as myocarditis, pericarditis, stress cardiomyopathy, etc., [18]. Patients without significant coronary artery disease are those whose major epicardial coronary arteries are either angiographically normal or with documented non-significant disease (<50% stenosis) by coronary angiogram [19].

### 2.2. Study Population

We identified all consecutive patients referred to our center during 2018 for cardiac catheterization due STEMI suspicion. Hospital Universitario Central de Asturias is a reference center for primary angioplasty and a national reference center for inherited cardiac conditions. Consecutive patients with confirmed premature atherothrombotic CAD presented as STEMI were recruited for “CholeSTEMI registry”. Premature CAD was defined according to DLCN criteria [7,8] as men <55 and women <60 years old. The existence of atherothrombotic CAD had to be confirmed by coronary angiogram. Therefore, only STEMI due to type 1 MI were recruited. Patients without significant coronary artery disease [19] and patients with STEMI due coronary artery dissection (type 2 MI) were excluded. Patients who died could not be included in the study.

### 2.3. Clinical Evaluation

We retrospectively collected clinical data from this cohort. Total cholesterol (TC), LDLc, high-density lipoprotein cholesterol (HDLc) and Triglycerides (TG) levels during admission for STEMI and previous statin treatment were reviewed. In addition, classical cardiovascular risk factors like high blood pressure, tobacco consumption, diabetes, dyslipidemia, and body mass index (BMI) were also collected. Historical cholesterol levels were also reviewed.

Prospectively, patients that met the inclusion criteria were contacted by phone during the early months of 2020 (January–February) and were offered them to participate in the study. Patients that agreed to participate in genetic testing were given an appointment so that the study could be explained with more details. Finally, 56 patients agreed to participate and signed the informed consent for genetic testing. Blood samples were collected both for genetic testing and for new cholesterol levels determination.

In this appointment DLCN criteria were evaluated. Patient’s family history of premature cardiovascular disease or hypercholesterolemia in their first-degree relatives as well as their own cardiovascular history and their untreated lipid levels were investigated. Moreover, physical signs such as the presence of tendon xanthomata or arcus cornealis were also evaluated. In patients without known hypercholesterolemia history and no previous available LDLc records, we considered the LDLc levels during STEMI admission. If there were previous records of LDLc determination, the highest level recorded before statin treatment was the one use for the score calculation. If the latest LDL available was already under lipid-lowering meditation, LDLc levels were estimated with correction factors considering the drug and its dose, as previously reported [20].

The CholeSTEMI study was approved by the local Ethical Committee (Principado de Asturias; registry number 2020/003).

### 2.4. Genetic Testing

We evaluated the 3 genes related to heterozygous FH (*LDLR*, *APOB*, *PCSK9*), one gene related to autosomal recessive FH (*LDLRAP1*) and 2 other genes related to hyperlipidemia (*APOE* and *LIPA*). Genetic screening was carried out with DNA samples from the recruited patients. Written informed consent was obtained from all participants prior to genetic study. All of them were NGS sequenced for a gene panel including the coding sequence plus at least 5 flanking intronic base pairs of *LDLR*, *APOB*, *PCSK9*, *APOE*, *LDLRAP1*, and *LIPA* genes by Ion Torrent semiconductor chip technology in a Ion GeneStudio S5 Sequencer (Thermo Fisher Scientific, Waltham, MA, SUA), according to previously described protocols [21,22]. (Appendix A).

Overall, in silico coverage of the familial hypercholesteromia and hyperlipidemia associated genes 100% (Appendix A). Variant Caller v5 software was used to variant identification (Thermo Fisher Scientific). Ion Reporter (Thermo Fisher Scientific, Waltham, MA, USA) and HD Genome One (DREAMgenics S.L., www.dreamgenics.com, Oviedo, Spain) softwares were used for variant annotation, including population, functional, disease-related and in silico predictive algorithms databases.

Interpretation of all gene variants with an allele frequency <0.01 was based the American College of Medical Genetics and Genomics (ACMG-AMP) 2015 Standards and Guidelines [23]. According to ACMG-AMP criteria, variants pathogenic (P), likely-pathogenic (LP), and variants of uncertain significance (VUS).

### 2.5. Statistical Analysis

Statistical analyses were performed with SPSS v.19. Descriptive data for continuous variables are presented as mean ± SD and as frequencies or percentages for categorical variables. The Chi-square test or Fisher exact test was used to compare frequencies whereas differences in continuous variables were evaluated with either the Student T test or Mann–Whitney U test. *p* < 0.05 was considered to be significant.

## 3. Results

In 2018, 365 consecutive patients were referred for cardiac catheterization to our center in 2018 due to STEMI suspicion. According to the inclusion criteria, 80 patients alive with confirmed diagnosis of premature atherothrombotic STEMI after coronary angiography were identified (Figure 1). Men <55 years old represented the 85% (68 male patients) of this initial cohort and woman <60 years old the 15% (12 female patients).

A good response rate was achieved: 56 patients out of 80 (70% response) agreed to participate in the study and underwent genetic screening. Gender distribution was similar than in the original population: 48 men and 8 women (85.7% male patients). Mean age was 46.4 years old for men (±6.9) and 54.75 for women (±3.4).

Prevalence of classical cardiovascular risk factors prior to STEMI admission is shown in Table 1. Smoking and overweight were the most prevalent risk factors in this population. Only 16% of the cohort had a normal BMI. Further, 82.1% of the cohort had a BMI > 25 (overweighed), and 41% of them were obese (BMI > 30). There were no significant differences between genders between smoking, dyslipidemia, hypertension, or diabetes (Table 1). Average BMI in women was slightly lower than in men (Table 1), but both were within the overweight range.

Moreover, about 32.1% of this population presented family history of premature CVD in first-degree relatives. Thus, if considered in second-degree relatives or up to 70 years old, this percentage would reach the 50% of our cohort. Only one male patient presented personal history of kidney failure. Moreover, there were also other relevant risk factors for premature CVD present in our population. For instance, between women, one presented hemochromatosis, one had a myeloproliferative syndrome and a third one was under oral contraceptives. Among men, one had been treated for lymphoma, two consumed cocaine, one was HIV positive, two had psoriasis, and another one had rheumatoid arthritis under treatment.

According to DLCN clinical criteria for FH diagnosis, no one reached a definite clinical diagnosis (≥8 points) and only one male patient reached a probable clinical diagnosis of FH (6–8 points). FH was possible in 51.8% of the cohort, whereas in 46.4% it was unlikely. In most women, FH was unlikely (62.5%) and only three of them reached three points in the score. Conversely, FH was possible in most men (54.2%). No women had known LDLc levels > 155 mg/dl and therefore, no woman score any point in due to LDLc according to DLCN criteria. However, 10 men (20.1% of them) had LDLc levels between 155–189 md/dL (scoring 1 point) and six men (12.5%) had 190–249 mg/dL (scoring three points). From these men, 25% of them were identified during STEMI hospitalization without prior LDLc known values. Nobody had LDLc > 250 mg/dL (5–8 points). Extensive clinical data and individual scores are available in Appendix A.

Although not significant differences between genders in cholesterol levels were found (Table 2), mean LDL levels analyzed during hospitalization for STEMI in 2018 were higher in men (112.9 ± 36.9 SD) than in women (94.75 ± 29.2 SD). At discharge, all patients were prescribed high doses of statins (atorvastatin 80 mg ± ezetimbe) and received lifestyle recommendations. Mean LDLc levels determined in 2020 after Atorvastatin 80 mg were expected to be reduced a 55%, reaching the goal of being under 55 mg/dl recommended in secondary prevention. They were reduced to lower levels in men (64.7 ± 31.9 SD) than in women (68.6 ± 31.2 SD), but unfortunately both mean values were above the recommended cut-off. There was a higher LDLc reduction in men (42.8%) than in women (27.6%).

We identified *LDLR* p.Asp221Gly (c.662A > G; NM_000527) pathogenic variant in a male of 44 years old (patient 8, Appendix A). He was the only patient with a probable DLCN score (6 points). His highest LDLc level recorded was 217. However, as he was under statin treatment, his LDL levels at STEMI were 82 mg/dL. Another male was carrier of *APOB* p.Phe3150Leu (c.9448T > C; NM_000527) variant of unknown significance (patient 5, Appendix A). The global genetic yield was 1.8%, with no other relevant genetic variants were identified.

## 4. Discussion

Several studies have shown that genetic testing for FH is currently underused, even in patients with known severe hypercholesterolemia [24,25]. Overall, it is estimated that less than 1% of the people with FH are properly diagnosed [14,26]. Despite many studies available have analyzed the prevalence of FH by clinical criteria, little is known about the prevalence of genetically confirmed FH, especially in patients with premature CAD [27]. It was believed that pathogenic FH variants in patients with premature CAD were 15–20 times more frequently found that in general population [16,27]. However, a recent meta-analysis in CAD showed that prevalence of FH varies widely between studies, from 0.4% to up to 25.4% [5]. Nonetheless, other interesting studies, as well as our results, showed a lower prevalence of FH in premature CAD than expected. Nanchen et al. showed that clinical FH prevalence (without genetic testing confirmation) in patients with acute coronary syndrome was 1.6%, increasing to 4.8% in those with premature CAD [28]. These differences can be explained by several factors. Firstly, in most studies the prevalence of FH is evaluated only by clinical criteria and these criteria have important limitations that may underestimate the score. For instance, obtaining previous lipid profile records or a proper history about first-degree relatives can be difficult. Moreover, the inclusion criteria are very variable between studies. On the one hand, the cut-off age for considering CAD “premature” is inconsistent. On the other hand, different kinds of CAD are included. In this sense, it must be highlighted that atherothrombotic CAD in type 1 MI has a completely different etiopathogenic mechanism from the type 2 MI secondary to anemia or coronary embolus. Therefore, properly identifing atherothrombotic MI and CAD is crucial. In fact, if CAD is absent, the benefits of cardiovascular risk reduction strategies with type 2 MI remain uncertain [17]. Most relevant studies exploring FH prevalence in premature CAD are shown in Table 3 [27,28,29,30,31,32,33,34,35,36].

An interesting Swedish study have tried to evaluate the prevalence of genetically confirmed FH in premature CAD [27]. After evaluating the three heterozygous FH-related genes (LDLR, APOB and PCSK9), they reported a genetically confirmed prevalence of heterozygous FH of 4.5%. However, this study had some limitations. Firstly, not all patients genetically tested had a final diagnosis of myocardial infarction (STEMI or NSTEMI) after coronary angiogram. In fact, only 62% of them had a definite diagnosed of myocardial infarction after cardiac catheterization. Moreover, they achieved a poor response rate to the study and only 30% of the patients who met the inclusion criteria were finally sequenced [27]. Conversely, the response rate in our cohort was 70%. Moreover, they had a different cut-off age for the inclusion criteria. They considered “premature CAD” patients 5 years younger than in DLCN criteria (age <50 years for men and <55 years for women [7,8]. Therefore, the slightly higher prevalence of FH found in the Swedish study (4.5%) compared to ours (1.8%), may be explained by these differences in the population age.

A recent excellent Chinese study, with a similar design to ours, included consecutive patients with premature (male ≤55 years old, and female ≤60 years old) myocardial infarction and analyzed the 4 genes (*LDLR*, *APOB*, *PCSK9* and *LDLRAP1*). The prevalence of FH diagnosed by genetic testing was also slightly higher than ours, a 4.4%. However, patients with incomplete clinical data or no blood samples were excluded without specifying the percentage of losses [30]. Moreover, MI definition not only included type 1 MI presenting with STEMI, but also other types of MI.

To our knowledge, we present the first study that systematically evaluated by genetic testing consecutive patients presenting with premature STEMI, with confirmed atherothrombotic CAD by coronary angiogram and using the same cut-off age to define premature CAD as in DLCN criteria (male <55 years old, and female <60 years old [7,8,10], irrespective of other clinical criteria. Patients were NGS sequenced for a gene panel including the three genes related to heterozygous FH (*LDLR*, *APOB*, *PCSK9)*, one related to autosomal recessive FH (*LDLRAP1*) and two other genes related to hyperlipidemia (*APOE* and *LIPA*). Another study that analyzed the prevalence of DNA-confirmed FH in a cohort 77 young patients (considered if ≤50 years old) with myocardial infarction did find a prevalence similar to ours (1.3% vs. 1.8%) [35].

Although it has been considered that genetic testing in general population is not cost-effective [24,25], some studies have suggested that FH screening programs could be useful in patients with premature CAD [12]. The consensus statement from the European Atherosclerosis considers that patients with premature CAD should be screened for FH [14]. Despite the prevalence of genetically confirmed FH in premature CAD in our cohort was 4.5–9 times higher than in general population [28], we believe that it is still too low to systematically recommend genetic screening in all premature CAD patients. Further screening programs in premature CAD should be organized combining clinical DLCN criteria and genetics, saving genetic testing for individuals with a probable to definite clinical FH diagnosis. Although some studies did not show clear correlation between DLCN criteria score and genetic testing results for FH [29], most of them did find similar data when evaluated by genetic testing or clinical criteria [5,16,27]. In fact, in our study, the clinical score did identify the only patient with a definite diagnosis of FH confirmed by DNA testing. What is more, it could be said that the genetic yield among patients with a probable to definite FH clinical diagnosis in our cohort reaches a 100%.

Apart from that, it is noticeable that in our cohort, the patient with definite FH confirmed by genetic testing was already under hypolipemiant treatment and his LDLc levels at STEMI were into normal limits. Therefore, when assessing DLCN criteria, previous LDLc must be carefully evaluated. Had LDLc previous records not been available, clinical FH criteria would have failed to identify him. Moreover, suffering STEMI with low LDL levels supports the idea that the presence of FH itself is more relevant than a single LDL determination [13,37,38], and that high intensive treatment should be promptly initiated. Moreover, genetic family screening for the pathogenic variant will allow us to identify all relatives at high CVD risk and to initiate intensive hypolipemiant treatment.

On the other hand, as in other similar studies [35] classical cardiovascular risk factors prevalence, specially smoking and elevated BMI, far exceeds the prevalence of FH in patients with premature STEMI. In our cohort, we did not find basal significant differences in classical CVD risk factors between genders. LDLc levels at STEMI were slightly higher in men than in women and all patients were prescribed high doses of statins and received lifestyle recommendations expecting a well-controlled LDLc levels in follow-up. However, LDLc control levels were out of range in both genders according to current dyslipidemia guidelines LDL target [10], and slightly worse in women (with only a 27.6% reduction). Optimizing long-term lipid treatment in all patients with premature atherothrombotic CAD is required. The availability of PCSK9-inhibitors provides a new clinical scenario, helping to dramatically reduce LDL-C to target levels [11]. Moreover, other cardiovascular risk factors in the young such as psoriasis and rheumatoid arthritis should also be considered.

We believe that the strength of our study relies in the homogeneity of the cohort, recruiting all consecutive patients with premature confirmed atherothrombotic CAD presented as STEMI and with a good compliance to the study, compared to other similar studies. We systematically evaluated the main genes associated to FH, regardless of their cholesterol levels or other clinical criteria. However, there were several limitations. This is a retrospective and observational study, with its inherent limitations. Patients were enrolled form a single center. Moreover, the sample size, although similar to other studies, is relatively small and, therefore, makes it difficult to draw definite conclusions.

## 5. Conclusions

The prevalence of FH in this specific cohort of patients with premature atherothrombotic CAD seems low (1.8%) but still 4.5–9 times higher than in general population. In order to improve the genetic yield, genetic testing may be considered among patients with probable or definitive FH diagnosis according to clinical criteria. The classical cardiovascular risk factors prevalence far exceeds FH prevalence in patients with premature STEMI. LDLc control levels after STEMI were out range, despite intensive treatment. These findings reinforce the need for more aggressive preventive strategies in the young and for intensive lipid-lowering therapy in secondary prevention.

## Figures and Tables

**Figure 1 jcm-09-03489-f001:**
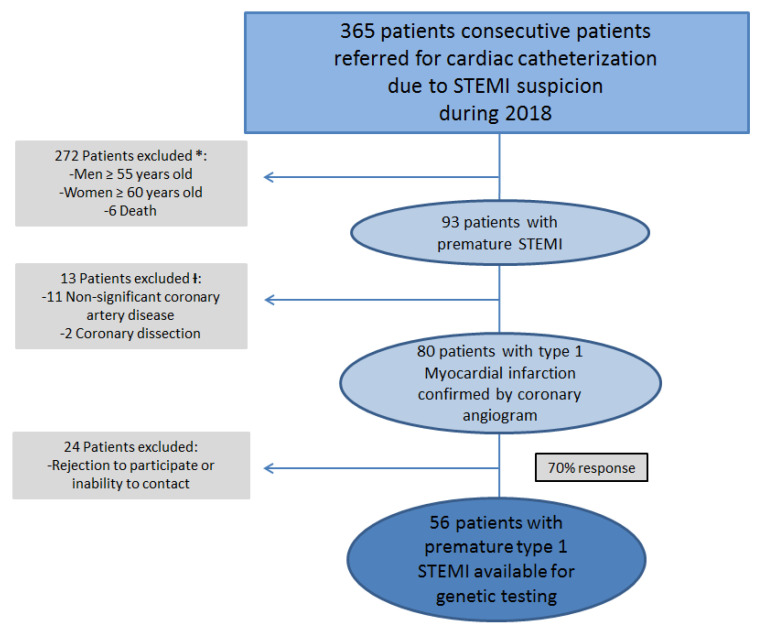
Patient selection. Flowchart showing steps for inclusion criteria. STEMI = Myocardial infarction with ST segment elevation; PCI = Primary percutaneous coronary intervention. * Premature CAD was defined according to DLCN criteria [7,8] as men <55 and women <60 years old. Ɨ Only STEMI due to type 1 MI confirmed by coronary angiogram were included.

**Table 1 jcm-09-03489-t001:** Known cardiovascular risk factors of the cohort after STEMI.

	Whole Cohort (*n* = 56)	Women (*n* = 8)	Men (*n* = 48)	*p*
Smoking	85.7%	100%	83.3%	0.21
Dyslipidemia	42.9%	37.5%	43.8%	0.74
Hypertension	28.6%	25%	29.2%	0.81
Diabetes	10.7%	12.5%	10.4%	0.86
Average BMI	28.9 ± 3.8 SD	28.5± 4.7 SD	29 ± 3.6 SD	0.7

BMI = body mass index.

**Table 2 jcm-09-03489-t002:** Cholesterol levels.

	Cohort at STEMI (*n* = 56)	Women at STEMI (*n* = 8)	Men at STEMI (*n* = 48)	Cohort at Control	*p*	Women at Control (*n* = 8)	Men at Control (*n* = 48)	*p*
Total cholesterol (mg/dL)	179.8 ± 39.3 SD	168.4 ± 29.4 SD	181.7 ± 40.6 SD	138.6 ± 50.4 SD	*p* > 0.05	137.4 ± 37.1 SD	138.8 ± 52.6 SD	*p* > 0.05
LDLc (mg/dL)	110.4 ± 39.3 SD	94.75 ± 29.2 SD	112.9 ± 36.9 SD	65.3 ± 31.5 SD	*p* > 0.05	68.6 ± 31.2 SD	64.7 ± 31.9 SD	*p* > 0.05
HDLc (mg/dL)	39.2 ± 12.2 SD	48.6 ± 18.6 SD	37.6 ± 10.3 SD	43.5 ± 16.8 SD	*p* > 0.05	44 ± 15.3 SD	43.1 ± 17.2 SD	*p* > 0.05
Triglycerides (mg/dL)	176.9 ± 117.8 SD	144.5 ± 104.2 SD	182.3 ± 120.1 SD	190.6 ± 393.5 SD	*p* > 0.05	140.9 ± 89.1 SD	199.2 ± 423.7 SD	*p* > 0.05

STEMI, ST-elevation myocardial infarction; LDLc, low-density lipoprotein cholesterol; HDLc, high-density lipoprotein cholesterol.

**Table 3 jcm-09-03489-t003:** Prevalence of familiar hypercholesterolemia in premature coronary artery disease according to different studies.

	N	Inclusion Criteria	Clinical Presentation of CAD	Clinical Diagnosis (DLCN Probable-Definite) (%)	Genetic Diagnosis (%)
Amor-Salamanca et al. [29]	103	≤65 years old and LDLc ≥160 mg/dL	STEMI, NSTEMI and unstable angina	27.2%	8.7%
Cui et al. [30]	225	≤55 years old in men and ≤60 in women	Myocardial infarction (STEMI and NSTEMI)	5.3%	4.4%
De Backer et al. [31], Subanalysis of premature CAD	2212	<60 years old	ACS, acute myocardial ischemia or procedure: coronary artery bypass grafting or percutaneous coronary intervention)	15.4%	Not evaluated
Koivisto et al. [32]	90	≤45 years old	Symptomatic CAD (myocardial infarction and effort-induced angina pectoris)	Not evaluated	9%
Li et al. [33], Subanalysis of premature CAD	889	≤55 years old in men and ≤60 in women	Myocardial infarction (STEMI and NSTEMI)	7.1%	Not evaluated
Nanchen et al. [28] Subanalysis of premature CAD	1451	<55 years old in men and <60 in women	ACS (not specified whether STEMI, NSTEMI or unstable angina)	4.8%	Not evaluated
Pang et al. [34]	175	<60 years old	ACS, myocardial revascularization and angina pectoris	14.3%	Not evaluated
Pirazzi et al. [27]	49	<50 years old in men and <55 in women	Myocardial infarction (STEMI or NSTEMI), unstable angina, stable angina or atherosclerosis, spasm angina and other diagnosis (non- atherosclerotic)	16.7%	6.1%
Wald et al. [35]	231	≤50 years old	STEMI and NSTEMI	Not evaluated	1.3%
Yudi et al. [36]	210	≤55 years old in men and ≤60 in women	ACS, angina, unstable angina, STEMI and NSTEMI	1.4%	Not evaluated
Lorca et al. (this study).	56	<55 years old in men and <60 females in women	STEMIConfirmed by coronary angiogram	1.8%	1.8%

ACS: acute coronary syndrome; CAD: coronary artery disease; DLCN: Dutch Lipid Clinic Network; STEMI: myocardial infarction with ST elevation; NSETMI: myocardial infarction without ST elevation.

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
