# Peer review of "Familial Hypercholesterolemia in Premature Acute Coronary Syndrome. Insights from CholeSTEMI Registry"

_jcm, 2020, doi:10.3390/jcm9113489_

Round 1

Reviewer 1 Report

The topic  is  relevant   for  management of patients  with  recent  myocardial infarction.

The  genetic detection of FH may  prompt  more  aggressive reduction of LDL  cholesterol. This  point  should be  addressed in the  discussion section (PCSK9i use, etc).

The paper  i well written:  the introduction  is , in my opinion,   too  long  and  may be  shortened.

The limitations  of DCLN   score  should be  underlined:  often it is very  difficult  to obtain  lipid  profile  and careful history  of  first -degree  relatives, and  the score may be underestimated.

There are several  papers on the prevalence of FH in CAD patients. A table  for  comparison of previous  studies  with the present  could  be  useful.

Author Response

The topic is relevant   for management of patients with recent myocardial infarction.

The genetic detection of FH may prompt more aggressive reduction of LDL cholesterol. This point should be addressed in the discussion section (PCSK9i use, etc).

We thank the referee for his/her comment and have addressed this point in the discussion section of the manuscript according to the feedback provided.

Line 305: “Optimizing long-term lipid treatment in all patients with premature atherothrombotic CAD is required. The availability of PCSK9-inhibitors provides a new clinical scenario, helping to dramatically reduce LDL-C to target levels”

The paper is well written:  the introduction is, in my opinion,   too long and may be shortened.

We thank the referee for his/her comment and have tried, as suggested, to shorten the introduction.  

The limitations  of DCLN   score  should be  underlined:  often it is very  difficult  to obtain  lipid  profile  and careful history  of  first -degree  relatives, and  the score may be underestimated.

We thank the reviewer for his/her comment. We totally agree with the point raised by the referee and have highlighted this in the discussion section.

Line 235: “FH diagnosis based on DCLN score presents important limitations that may underestimate the score, as it is often it is very difficult to obtain lipid profile and careful history of first -degree relatives”.

There are several papers on the prevalence of FH in CAD patients. A table for comparison of previous studies with the present could be useful.

We thank the reviewer for his/her comment. We have added a table for comparison as suggested (Table 3), where the variability of inclusion criteria both in age and diagnostic criteria can be easily compared.

Reviewer 2 Report

In the present manuscript the authors aimed to systematically evaluate the prevalence of FH evaluated by both clinical and genetic testing, in a cohort of confirmed premature CAD patients presenting with myocardial infarction with ST elevation (STEMI). The authors also evaluated known cardiovascular risk factors, cholesterol levels (at STEMI, prior to STEMI and after STEMI) and differences between genders in this cohort. Overall, 365 consecutive patients, who were referred for cardiac catheterization and 80 patients of those with premature CAD, confirmed by coronary angiography included in this registry. Only 56 patients were available for the genetic testing. According to DLCN clinical criteria for FH diagnosis, no patient reached a definite clinical diagnosis (>8 points) and only one male patient reached a probable clinical diagnosis of FH (6-8 points).

The following comments apply:

Major comments:

  • The design of the study is not clear. What was the purpose of the Register? Did the patients sign the informed consent for the registry?
  • Page 3/Line 120: It seems to be the ethical Approvel for the genetic analysis. Could you provide the ethical Approvel register?
  • Thersholds for premature CAD in female patients is not in accordance to the usually used cutoff
  • No general population cohort is added for control
  • If the aim was to determine the prevalence of premature CAD, the study is far to small in terms of sample size to provide any significant power
  • It is unclear whether premature CAD refers to STEMIs only or also for non-acute CAD manifestations
  • Females are underrepresentated in this cohort (n=12)
  • Statistical power is not sufficient for comparison of risk factors between men and women
  • How is the cohort at control defined. Please provide n for each group in each table.
  • Conclusions that can be drawn from the present analysis are hypothetical, given the study design and small cohort.

Author Response

In the present manuscript the authors aimed to systematically evaluate the prevalence of FH evaluated by both clinical and genetic testing, in a cohort of confirmed premature CAD patients presenting with myocardial infarction with ST elevation (STEMI). The authors also evaluated known cardiovascular risk factors, cholesterol levels (at STEMI, prior to STEMI and after STEMI) and differences between genders in this cohort. Overall, 365 consecutive patients, who were referred for cardiac catheterization and 80 patients of those with premature CAD, confirmed by coronary angiography included in this registry. Only 56 patients were available for the genetic testing. According to DLCN clinical criteria for FH diagnosis, no patient reached a definite clinical diagnosis (>8 points) and only one male patient reached a probable clinical diagnosis of FH (6-8 points).

The following comments apply:

Major comments:

We thank the reviewer for his/her valuable comments that have helped to improve the quality of the manuscript.

In order to try to answer the main concerns about the characteristic of our cohort, we have added a new section in Methods to clarify the definitions of “acute coronary syndrome”, “myocardial infarction” and “STEMI”.

We hope this may help to highlight that, unlike other studies including different kinds of myocardial infarction (that contained patients without significant CAD), we were able to recruit a homogenous cohort of patients with a definite diagnosis atherothrombotic CAD presenting as STEMI.

"2.1. General definitions

There are 3 traditional types of acute coronary syndrome (ACS): ST-elevation myocardial infarction (STEMI), Non-ST-elevation myocardial infarction (NSTEMI) and unstable angina. Acute myocardial infarction (MI) is defined by acute myocardial injury detected by abnormal cardiac biomarkers in the setting of evidence of acute myocardial ischaemia (17). MI is classified in 5 types, based on the etiology and pathogenesis (17). MI caused by atherothrombotic CAD and usually precipitated by atherosclerotic plaque disruption is designated as a type 1 MI. However, in type 2 MI there is demand-supply mismatch resulting in myocardial ischemia (17). This demand supply mismatch can be due to multiple reasons including but not limited to presence coronary obstruction. Mechanisms include hypoxemia hypo/hypertension, arrhythmias, anemia, and also vasospasm, acute aortic dissection coronary embolus, spontaneous coronary artery dissection, etc (17).

Patients with acute chest pain and persistent ST-segment elevation on a surface electrocardiogram (generally reflects an acute total or subtotal coronary occlusion) ultimately develop STEMI (18). Differential Diagnosis includes other pathologies that can cause ST-segment elevations without significant coronary artery disesase, such as myocarditis, pericarditis, stress cardiomyopathy, etc (18). Patients without significant coronary artery disease are those whose major epicardial coronary arteries are either angiographically normal or with documented non-significant disease (< 50% stenosis) by coronary angiogram (19)".

Moreover, following reviewer 1 advice, we have added a table (Table 3), where the variability of inclusion criteria in other studies both in age and diagnostic criteria can be easily compared.

“Inclusion criteria are very variable not only in terms of age but also in types of included CAD. Atherothrombotic CAD in type 1 MI has a completely different etiopathogenic mechanism to type 2 MI secondary to anemia or coronary embolus. Even general cardiology studies have shown variable occurrences of type 2 MI depending on criteria used for diagnosis (17). In fact, if CAD is absent, the benefits of cardiovascular risk reduction strategies with type 2 MI remain uncertain (17).

  1. Thygesen K, Alpert JS, Jaffe AS, Chaitman BR, Bax JJ, Morrow DA, et al. Fourth universal definition of myocardial infarction (2018). European Heart Journal. 2019 Jan 14;40(3):237–69.
  2. Collet J-P, Thiele H, Barbato E, Barthélémy O, Bauersachs J, Bhatt DL, et al. 2020 ESC Guidelines for the management of acute coronary syndromes in patients presenting without persistent ST-segment elevation. European Heart Journal. 2020 Aug 29;ehaa575.
  3. Cannon RO. Chest Pain with Normal Coronary Angiograms. N Engl J Med. 1993 Jun 10;328(23):1706–8.

The design of the study is not clear. What was the purpose of the Register? Did the patients sign the informed consent for the registry?

We thank the reviewer for this comment and have tried to clarify the purpose of this study. Page 1, line 86:  “In this study, we aimed to explore the prevalence of a genetic condition (FH) within a selected population with confirmed atherothrombotic CAD”.

Page 2, line 87, continued: “we systematically evaluated the prevalence of FH evaluated by both clinical and genetic testing, in a cohort of confirmed premature atherothrombotic CAD patients presenting with myocardial infarction with ST elevation (STEMI)”

Secondly, as explained in Section 2.3.Clinical evaluation, line 131, “56 patients agreed to participate and signed the informed consent for genetic testing”.

Page 3/Line 120: It seems to be the ethical Approvel for the genetic analysis. Could you provide the ethical Approvel register?

We thank the reviewer for highlighting the importance of the ethical Approval register. The number was specified in banquets. “The CholeSTEMI study was approved by the local Ethical Committee (Principado de Asturias; registry number 2020/003)”.

Thresholds for premature CAD in female patients is not in accordance to the usually used cutoff.

We thank the reviewer for his/her comment and we totally agree with the variability of cutoff used between previous studies. This has been highlighted in new Table 3.

Therefore, we had decided to use “the same cut-off age to define premature CAD as in DLCN criteria (male <55 years old, and female <60 years old, (7-8, 10))”, line 252.

  1. Civeira F. Guidelines for the diagnosis and management of heterozygous familial hypercholesterolemia. Atherosclerosis. 2004 Mar;173(1):55–68.
  2. WHO Human Genetics Programme. (‎1999)‎. Familial hypercholesterolaemia (‎‎‎‎FH)‎‎‎‎ : report of a second WHO consultation, Geneva, 4 September 1998. World Health Organization.
  3. Mach F, Baigent C, Catapano AL, Koskinas KC, Casula M, Badimon L, et al. 2019 ESC/EAS Guidelines for the management of dyslipidaemias: lipid modification to reduce cardiovascular risk. European Heart Journal. 2020 Jan 1;41(1):111–88.

No general population cohort is added for control.

We agree with the reviewer. Due to the fact the purpose of the study was explore the prevalence of FH within a selected population, it was not designed with a control cohort.

If the aim was to determine the prevalence of premature CAD, the study is far to small in terms of sample size to provide any significant power.

We apologize by the misunderstanding. The purpose of this study was not to determine the prevalence of premature CAD, but to explore the “prevalence of FH” in a selected population of patients with premature CAD.

As it was designed as a retrospective study to evaluate the prevalence of FH in a limited selected population, no sample size calculations to estimate the power were performed. However, we agree about the relatively small simple size.

Therefore, we have highlighted it in the limitations section. “The sample size, although similar to other studies, is relatively small and, therefore, makes it difficult to draw definite conclusions”, line 322.

Moreover, due to this limitation, we have rephrased the conclusion to express a hypothesis with date from a limited population. “The prevalence of FH in this specific cohort of patients with premature atherothrombotic CAD seems low”. “In order to improve the genetic yield, genetic testing may be considered among patients with probable or definitive FH diagnosis according to clinical criteria”, line 326.

It is unclear whether premature CAD refers to STEMIs only or also for non-acute CAD manifestations.

We thank the reviewer for his/her comment.

It has now been clarified (Section 2.1, line 95) that all STEMI are considered by definition ACS.

Therefore our cohort included all patients from 2018 with ACS presenting as STEMI, with confirmed atherothrombotic CAD by coronary angiogram (myocardial infarction type 1).

Females are underrepresented in this cohort (n=12).

We agree with the reviewer for his/her comment. Women seem to be always underrepresented not only in premature cardiovascular disease but also in all-ages STEMI compared to men.

-Collet J-P, Thiele H, Barbato E, Barthélémy O, Bauersachs J, Bhatt DL, et al. 2020 ESC Guidelines for the management of acute coronary syndromes in patients presenting without persistent ST-segment elevation. European Heart Journal. 2020 Aug 29;ehaa575.

Statistical power is not sufficient for comparison of risk factors between men and women.

We agree with the reviewer at this point and thank him/her for raising this concern.

This was not the purpose of the study and we only wanted to  give a visual descriptive comparison about classical risk factors, in order  to emphasize its high prevalence, but without further claims.

How is the cohort at control defined. Please provide n for each group in each table.

We thank the reviewer for his/her comment.

“Control cholesterol levels” were determined at the appointment in which they agreed to participate in the study.

Lines 132 “Blood samples were collected for genetic testing and new cholesterol levels determination”.

Moreover, we have added “n” for each group in each table (table 1 and table 2), as suggested.

Conclusions that can be drawn from the present analysis are hypothetical, given the study design and small cohort.

We totally agree with the reviewer. Therefore, we have modified the conclusions and limitation as explained previously.

Limitations: “The sample size, although similar to other studies, is relatively small and, therefore, makes it difficult to draw definite conclusions”.

We have rephrased the conclusion to express a hypothesis with date from a limited population: “The prevalence of FH in this specific cohort of patients with premature atherothrombotic CAD seems low”. “In order to improve the genetic yield, genetic testing may be considered among patients with probable or definitive FH diagnosis according to clinical criteria”.

Reviewer 3 Report

In this article the authors explore the prevalence of genetic FH in a small sample of patients qith a confirmed STEMI diagnosis. 

Despite dealing with an interesting still unsolved question explored already in other cohorts, the article presents some major criticisms and need to be thoroughly revised, as it is not acceptable in the present form.

The most important is the small sample size: was a power sample calculation performed? If the aim is the exploration of the prevalence of a genetic condition within a selected population, this should be performed and added in the methods section. The sample of the population is too small to draw conclusions on either the prevalence of FH in acute STEMI or the necessity of performing genetic testing. 

Here are some questions and suggestions.

Would you explain the reason for selectively choosing STEMI among ACS events? Why did you decide to exclude patients with non significant CAD or with coronary arteries dissection ? Please also clarify what you mean for non significant CAD.

In the evaluation of events, it would have been interesting to evaluate on a wider range all confirmed ACS.

The overall population had near-normal LDL-C levels. Did you have the opportunity to also dose Lp(a)?

Page 2, line 50 « leadS » ; line 55 « conditioN » ; line 60 « leads » ; line 72 « lowerED » ; line 78 « mediCation » ; line 93 « due TO » ;

Page 3, line 103 : please delete « dyslipidemia » as a cardiovascular risk factor ; line 117 « useD » ;

Page 7, line 260: "absenCE"

Rephrase page 2 line 60 : clinical diagnosis and molecular diagnosis are separate diagnostic strategies ;

Rephrase page 2 line 73-75 (not clear)

Author Response

In this article the authors explore the prevalence of genetic FH in a small sample of patients with a confirmed STEMI diagnosis.

Despite dealing with an interesting still unsolved question explored already in other cohorts, the article presents some major criticisms and need to be thoroughly revised, as it is not acceptable in the present form.

The most important is the small sample size: was a power sample calculation performed? If the aim is the exploration of the prevalence of a genetic condition within a selected population, this should be performed and added in the methods section. The sample of the population is too small to draw conclusions on either the prevalence of FH in acute STEMI or the necessity of performing genetic testing.

We thank the reviewer for his/her comment and suggestions that have helped to improve the quality of the manuscript.

We have rephrased the purpose of this study. Page 1, line 86:  “In this study, we aimed to explore the prevalence of a genetic condition (FH) within a selected population with confirmed atherothrombotic CAD”.

As it was designed as a retrospective study to evaluate the prevalence of FH in a limited selected population, no sample size calculations to estimate the power were performed. However, we agree about the relatively small simple size.

Therefore, we have highlighted it in the limitations section. “The sample size, although similar to other studies, is relatively small and, therefore, makes it difficult to draw definite conclusions”.

Moreover, due to this limitation, we have rephrased the conclusion to express a hypothesis with date from a limited population. “The prevalence of FH in this specific cohort of patients with premature atherothrombotic CAD seems low”. “In order to improve the genetic yield, genetic testing may be considered among patients with probable or definitive FH diagnosis according to clinical criteria”.

Here are some questions and suggestions.

Would you explain the reason for selectively choosing STEMI among ACS events?

Why did you decide to exclude patients with non significant CAD or with coronary arteries dissection ? Please also clarify what you mean for non significant CAD.

In the evaluation of events, it would have been interesting to evaluate on a wider range all confirmed ACS.

We thank again the reviewer for his/her comments and suggestions that have helped to improve the quality of the manuscript. In order to try to answer the main concerns about the characteristic of our cohort, we have added a new section in Methods to clarify the definitions of “acute coronary syndrome”, “myocardial infarction” and “STEMI”. We hope this may help to highlight that, unlike other studies including different kinds of myocardial infarction (that contained patients without significant CAD), we were able to recruit a homogenous cohort of patients with a definite diagnosis atherothrombotic CAD presenting as STEMI.

"2.1. General definitions

There are 3 traditional types of acute coronary syndrome (ACS): ST-elevation myocardial infarction (STEMI), Non-ST-elevation myocardial infarction (NSTEMI) and unstable angina. Acute myocardial infarction (MI) is defined by acute myocardial injury detected by abnormal cardiac biomarkers in the setting of evidence of acute myocardial ischaemia (17). MI is classified in 5 types, based on the etiology and pathogenesis (17). MI caused by atherothrombotic CAD and usually precipitated by atherosclerotic plaque disruption is designated as a type 1 MI. However, in type 2 MI there is demand-supply mismatch resulting in myocardial ischemia (17). This demand supply mismatch can be due to multiple reasons including but not limited to presence coronary obstruction. Mechanisms include hypoxemia hypo/hypertension, arrhythmias, anemia, and also vasospasm, acute aortic dissection coronary embolus, spontaneous coronary artery dissection, etc (17).

Patients with acute chest pain and persistent ST-segment elevation on a surface electrocardiogram (generally reflects an acute total or subtotal coronary occlusion) ultimately develop STEMI (18). Differential Diagnosis includes other pathologies that can cause ST-segment elevations without significant coronary artery disesase, such as myocarditis, pericarditis, stress cardiomyopathy, etc (18). Patients without significant coronary artery disease are those whose major epicardial coronary arteries are either angiographically normal or with documented non-significant disease (< 50% stenosis) by coronary angiogram (19).

Moreover, following reviewer 1 advice, we have added a table (Table 3), where the variability of inclusion criteria in other studies both in age and diagnostic criteria can be easily compared.

“Inclusion criteria are very variable not only in terms of age but also in types of included CAD. Atherothrombotic CAD in type 1 MI has a completely different etiopathogenic mechanism to type 2 MI secondary to anemia or coronary embolus. Even general cardiology studies have shown variable occurrences of type 2 MI depending on criteria used for diagnosis (17). In fact, if CAD is absent, the benefits of cardiovascular risk reduction strategies with type 2 MI remain uncertain (17).

  1. Thygesen K, Alpert JS, Jaffe AS, Chaitman BR, Bax JJ, Morrow DA, et al. Fourth universal definition of myocardial infarction (2018). European Heart Journal. 2019 Jan 14;40(3):237–69.
  2. Collet J-P, Thiele H, Barbato E, Barthélémy O, Bauersachs J, Bhatt DL, et al. 2020 ESC Guidelines for the management of acute coronary syndromes in patients presenting without persistent ST-segment elevation. European Heart Journal. 2020 Aug 29;ehaa575.
  3. Cannon RO. Chest Pain with Normal Coronary Angiograms. N Engl J Med. 1993 Jun 10;328(23):1706–8.

The overall population had near-normal LDL-C levels. Did you have the opportunity to also dose Lp(a)?

We thank the reviewer for his/her comment and totally agree about the interest in Lp(a) determination. However, this work has been supported by a grant (SECACCL-INV-CLI 20/003) whose badged only covered the genetic testing. Unfortunately, there were no more funding available for Lp(a) determination.

However, we really appreciate the suggestion and would really consider including this determination for further projects.

Round 2

Reviewer 2 Report

Thank you for your response. 

Author Response

Comments and Suggestions for Authors: Thank you for your response. 

We thank again the reviewer for his/her valuable comments that have helped to improve the quality of the manuscript.

Although there are no specific comments or suggestions to address in Round 2, taking advantage of the English editing asked, we have made an extra effort to try to improve the methods description and to present the results and discussion more clear.

Reviewer 3 Report

The authors have answered the questions and suggestions raised at the previous review. The manuscript has overall improved its quality. Some extensive english language editing is still required. Figure 1 could be adapted to the novel nomenclature based on recent guidelines in order to make it easier for the reader to go through the methods and population selection.

Author Response

Comments and Suggestions for Authors

The authors have answered the questions and suggestions raised at the previous review. The manuscript has overall improved its quality. Some extensive english language editing is still required. Figure 1 could be adapted to the novel nomenclature based on recent guidelines in order to make it easier for the reader to go through the methods and population selection.

We highly appreciate the reviewer valuable comments that have helped to improve the quality of the manuscript.

Figure 1 has been adapted as suggested.

Moreover, taking advantage of the English editing asked, we have made an extra effort to try to present the results and discussion more clear.